# Characterization of an MLP Homologue from *Haemaphysalis longicornis* (Acari: Ixodidae) Ticks

**DOI:** 10.3390/pathogens9040284

**Published:** 2020-04-14

**Authors:** Jin Luo, Hui Shen, Qiaoyun Ren, Guiquan Guan, Bo Zhao, Hong Yin, Ronggui Chen, Hongying Zhao, Jianxun Luo, Xiangrui Li, Guangyuan Liu

**Affiliations:** 1State Key Laboratory of Veterinary Etiological Biology, Key Laboratory of Veterinary Parasitology of Gansu Province, Lanzhou Veterinary Research Institute, Chinese Academy of Agricultural Science, Xujiaping 1, Lanzhou 730046, China; luojin02@caas.cn (J.L.); articks@126.com (H.S.); renqiaoyun@caas.cn (Q.R.); guanguiquan@caas.cn (G.G.); yinhong@caas.cn (H.Y.); luojianxun@caas.cn (J.L.); 2MOE Joint International Research Laboratory of Animal Health and Food Safety, College of Veterinary Medicine, Nanjing Agricultural University, Nanjing 210095, China; 3Gansu Agriculture Technology College, Duanjiatan 425, Lanzhou 730030, China; tsunami_zb@163.com; 4Jiangsu Co-Innovation Center for the Prevention and Control of Important Animal Infectious Disease and Zoonose, Yangzhou University, Yangzhou 225009, China; 5Ili Center of Animal Disease Control and Diagnosis, Ili 835000, China; chenronggui123@126.com; 6Chapchal Sibo Autonomous County Animal Husbandry and Veterinary Station, Chapchal 835400, China; zhaohy23@126.com

**Keywords:** ticks, *Haemaphysalis longicornis*, cysteine-rich protein, MLP

## Abstract

Members of the cysteine-rich protein (CRP) family are known to participate in muscle development in vertebrates. Muscle LIM protein (MLP) belongs to the CRP family and has an important function in the differentiation and proliferation of muscle cells. In this study, the full-length cDNA encoding MLP from *Haemaphysalis longicornis* (*H. longicornis*; HLMLP) ticks was obtained by 5′ rapid amplification of cDNA ends (RACE). To verify the transcriptional status of MLP in ticks, HLMLP gene expression was assessed during various developmental stages by real-time PCR (RT-PCR). Interestingly, HLMLP expression in the integument was significantly (*P* < 0.01) higher than that observed in other tested tissues of engorged adult ticks. In addition, HLMLP mRNA levels were significantly downregulated in response to thermal stress at 4 °C for 48 h. Furthermore, recombinant HLMLP was expressed in *Escherichia coli*, and Western blot analysis showed that rabbit antiserum against *H. longicornis* adults recognized HLMLP and MLPs from different ticks. Ten 3-month-old rabbits that had never been exposed to ticks were used for the immunization and challenge experiments. The rabbits were divided into two groups of five rabbits each, where rabbits in the first group were immunized with HLMLP, while those in the second group were immunized with phosphate-buffered saline (PBS) diluent as controls. The vaccination of rabbits with the recombinant HLMLP conferred partial protective immunity against ticks, resulting in 20.00% mortality and a 17.44% reduction in the engorgement weight of adult ticks. These results suggest that HLMLP is not ideal as a candidate for use in anti-tick vaccines. However, the results of this study generated novel information on the MLP gene in *H. longicornis* and provide a basis for further investigation of the function of this gene that could potentially lead to a better understanding of the mechanism of myofiber determination and transformation

## 1. Introduction

The LIM domain is a modular protein motif that is present in single or multiple copies in a wide variety of eukaryotic proteins and is involved in regulating gene expression and cell differentiation during development [1,2,3]. Cysteine-rich proteins (CRPs) are evolutionarily conserved proteins that are involved in the regulation of muscle development and are linked to short repeat sequences rich in glycine [4,5,6]. Muscle LIM protein (MLP) is a member of the CRP family of LIM proteins that is required for muscle differentiation [7].

Genetic ablation of mouse MLP and mutations in human MLP result in dilated cardiomyopathy and cardiac hypertrophy, which indicates the crucial importance of this protein in maintaining normal cardiac function [8,9]. Furthermore, MLP (Mlp84B) loss leads to developmental arrest at the pupal stage in *Drosophila*, and the muscle-dependent morphogenetic movements necessary for pupation are severely compromised in Mlp84B mutants [10].

*Haemaphysalis longicornis* ticks progress through four stages of life, the egg, larva, nymph, and adult stages, through their complete life cycle. Muscle performs necessary functions during blood sucking by moving the coxae of the appendages, retracting the chelicerae, and controlling pharyngeal action [11]. In addition, *Drosophila* Mlp84B can cooperate with D-titin to maintain muscle structural integrity [12]. Furthermore, as blood sucking leads to the stretching of muscle myofilaments [13], MLP may be essential in muscles under this increased pressure. Previous studies have identified muscle-associated molecules, such as actin, myosin alkali light chain, paramyosin, and troponin I as vaccine candidates for inducing protective immunity against ticks [14,15,16,17].

However, no MLP-encoding gene has been reported in arthropods to date. Thus, the objectives of the present study were to clone and characterize a cDNA encoding MLP from the tick *H. longicornis* and to evaluate the anti-tick immune effect of *H. longicornis* MLP in a rabbit model.

## 2. Materials and Methods

### 2.1. Ethics Approval

The present study was approved by the Ethics Committee of Lanzhou Veterinary Research Institute, Chinese Academy of Agricultural Sciences (approval no. LVRIAEC 2011-006), and the *H. longicornis* samples were collected in strict accordance with the requirements of the Ethics Procedures and Guidelines of the People’s Republic of China.

### 2.2. Ticks and Tissue Collection

Adult *H. longicornis*, *Haemaphysalis qinghaiensis*, *Hyalomma anatolicum*, *Hyalomma rufipes*, *Boophilus* (*Rhipicephalus*) *microplus*, and *Dermacentor silvarum* ticks were reared for several generations in a cloth bag attached to the back of a rabbit. The ticks were maintained at a temperature of 30 ± 2 °C and a relative humidity of 80% ± 5% through the different developmental stages. For tissue collection, *H. longicornis* adults were fed for 4 days in the bag on the rabbit back [18]. Subsequently, the midgut, ovaries, salivary glands, and integument were immediately transferred to phosphate-buffered saline (PBS) and washed three times, and the clean tissues were processed with TRIzol RNA extraction reagent (Invitrogen, China) and stored at −80 °C for later use.

### 2.3. Cloning and Sequencing the Full-Length cDNA of MLP

HLMLP was identified from expressed sequence tags (ESTs) constructed from a cDNA library of unfed female *H. longicornis* ticks, as described previously [19]. The full-length HLMLP cDNA of *H. longicornis* was obtained using a 5′ rapid amplification of cDNA ends (RACE) system (TaKaRa, Dalian, China) according to the manufacturer’s instructions. A gene-specific primer (GSP: TGCTCATGGCGCACTCCGTGTTG) was designed from the known 3′ fragment and used in 5′ RACE to amplify and clone the full-length HLMLP cDNA. The full-length sequences of HqMLP, HaMLP, HrMLP, BmMLP, and DsMLP were subsequently PCR-amplified from their respective cDNAs with the universal primers MLP-F (5′-ATGCCTTTCAAGCCCGT-3′) and MLP-R (5′-TTAGCCGTAGGTRGGGTCGTG-3′). The primers used in this study were synthesized by TaKaRa, Dalian, China. The PCR products were purified using a TaKaRa Agarose Gel DNA Purification Kit Ver. 2.0 (TaKaRa, Dalian, China), and the amplified products were ligated into the vector pMD^®^19-T (TaKaRa, Dalian, China)). The positive clones were sequenced with vector-specific primers (T7 and SP6) by Sangon (Shanghai, China). All sequences have been submitted to GenBank and can be retrieved with the accession numbers shown in Table 1.

### 2.4. Sequence Analysis

For homology analyses, the nucleotide and amino acid sequences were searched using the National Center for Biotechnology Information (NCBI) Basic Local Alignment Search Tool (BLAST) algorithm (http://www.ncbi.nlm.nih.gov/BLAST/). The Expert Protein Analysis System (http://us.expasy.org/) was used to deduce the amino acid sequence of HLMLP. PeptideMass (http://us.expasy.org/tools/peptide-mass.html) was used to determine the molecular weight and isoelectric point (pI) of HLMLP, and conserved domains were identified and analyzed using the Conserved Domain Database (CDD; http://www.ncbi.nlm.nih.gov/cdd). In addition, other motifs were identified using the Motif Scan program (http://hits.isb-sib.ch/cgi-bin/PFSCAN). The subcellular localization sites of proteins were predicted based on their amino acid sequences using WoLF PSORT (http://wolfpsort.org/). The deduced primary protein sequence of HLMLP was submitted to the SWISS-MODEL server to construct its three-dimensional (3D) model structure (http://swissmodel.expasy.org/), which was based on the 95% highest similarity and known 3D structure of chicken CRP1 chain A (PDB ID: 1b8t). The electrostatic surface potential of HLMLP was calculated using Swiss-PdbViewer v 4.1 (https://spdbv.vital-it.ch/).

### 2.5. Phylogenetic Analysis

MLPs from *H. longicornis* were aligned with previously identified MLP sequences. Nucleotide sequences and amino acid sequences from other species retrieved from NCBI GenBank were aligned using Clustal version 1.81. A neighbor-joining (NJ) phylogenetic tree was constructed using Molecular Evolutionary Genetics Analysis (MEGA) version 4.0 [20], and the reliability of the branching was tested using bootstrap re-sampling (1000 pseudo-replicates).

### 2.6. Expression of Recombinant MLP

The MLP sequence was PCR amplified using cDNA from *H. longicornis* adult ticks as template and GSPs (sense primer: 5′-CGGGATCCATGCCTTTCAAGCCCGT-3′, the shaded sequence indicates the ATG translation start codon and the underlined sequence indicates a *BamH*I restriction enzyme site; antisense primer: 5′-GGAATTCTTAGCCGTAGGTAGGGTCGTG-3′, an *EcoR*I restriction enzyme site is indicted by the underlined sequence). Then, the PCR product was ligated into the vector pGEM-T Easy. The purified recombinant plasmid was digested with *BamH*I/*EcoR*I restriction enzymes to create an insertion point, which was subsequently ligated to the *BamH*I/*EcoR*I cloning site of the pGEX-4T-1 expression vector. The resulting positive plasmids were transformed into the *E. coli* strain BL21 (DE3) competent cells (TaKaRa). To induce recombinant protein expression, a transformant was cultured in 10 mL of 2 × YT medium supplemented with a 1/1000 volume isopropyl-β-D-thiogalactoside (IPTG) (at a concentration of 1.0 mM) for 8 h at 37 °C and with shaking at 180 rpm. The recombinant rMLP was purified using the MagneGST™ Protein Purification System according to the manufacturer’s instructions (Promega, Madison, WI, USA), and the N- or C-terminal glutathione S-transferase (GST) tag was also induced and purified as a control under the same conditions. The purified protein and vector control were identified by sodium dodecyl sulphate–polyacrylamide gel electrophoresis (SDS-PAGE).

### 2.7. Immunoblot Analysis

Based on a previously described assay [21], immunoblot analysis was performed by subjecting rHLMLP and control protein to SDS-PAGE and then transferring the proteins onto nitrocellulose membranes, which were then incubated for 1 h with 5% skim milk powder. To analyze the reactivity of rHLMLP, rabbit antiserum generated against unfed adults and non-immune rabbit serum (as a negative control) were used at a dilution of 1:100 in Tris-buffered saline-Tween (TBST). Subsequently, the membranes were washed with TBST (pH 7.2) for ten minutes and then incubated with alkaline phosphatase-conjugated goat anti-rabbit IgG (Novagen, Madison, WI, USA) as a secondary antibody (diluted to 1:10,000). After the membranes were washed with TBST, the proteins that bound to the secondary antibody were visualized with nitroblue tetrazolium/5-bromo-4-chloro-3-indolyl phosphate (NBT/BCIP; Promega, Madison, WI, USA).

### 2.8. Temperature Treatments

*H. longicornis* eggs were collected in a polystyrene tube and placed at 4 °C for 48 h. Then, half of the chilled eggs were left at 4 °C, while the other half were heat shocked at 25 °C for 2 h. All eggs were prepared for cDNA synthesis based on total RNA isolated from *H. longicornis* eggs using a RevertAid First Strand cDNA Synthesis Kit (Fermentas, Shanghai, China) and stored at −20 °C for later use.

### 2.9. RT-PCR Analysis of HLMLP Expression in Tissues at Different Developmental Stages and Blood during Feeding

HLMLP expression levels were determined for different tissues and developmental stages in the ticks. The induction of HLMP transcription was determined during blood feeding, and the effects of cold and heat treatments on HLMLP from eggs were evaluated. Total RNA was extracted from eggs, unfed larvae, unfed nymphs, unfed females, and engorged females, as well as from various tissues, including the salivary glands, midgut, ovaries, and integument extracted from engorged female ticks. All RNA samples were used to synthesize single-strand cDNA from approximately 5 μg of total RNA using a RevertAid First Strand cDNA Synthesis kit (Fermentas, EU) and diluted 10-fold, with filtered distilled water used as a negative control template. HLMLP expression was detected by RT-PCR. Two GSPs (sense primer: 5′-AGGTTACGGCTTCGGTGGT-3′; antisense primer: 5′-TAGGGTCGTGTGGCTTGTTG-3′) were designed to amplify a 102 bp product, and sequencing of the PCR products was performed to determine the correctness of the product sequence. The primers *β*-actin F (5′-CGTTCCTGGGTATGGAATCG-3′) and *β*-actin R (5′-TCCACGTCGCACTTCATGAT-3′) were used to amplify a 69 bp fragment as a reference gene. The reaction conditions followed the instructions provided with the SYBR Premix ExTaq^TM^ II Kit (TaKaRa), and the PCR products were analyzed on 1% agarose gels.

### 2.10. Immunization and Challenge Infestation

Six New Zealand white rabbits (three months old) were purchased from a professional supplier and raised under tick-free conditions for use in the vaccine challenge experiment. The rabbits were divided into two groups of three rabbits that were either immunized with rHLMLP or injected with PBS as a control. A mixture of 0.5 mg of recombinant protein or PBS (pH of 7.5) emulsified in Freund’s complete adjuvant was subcutaneously inoculated into each rabbit. The same method and amount of purified antigen or PBS (pH of 7.5) was administered into each rabbit 30 days after the first inoculation. Approximately 30 days after the second inoculation, the purified antigen or PBS was emulsified in Freund’s incomplete adjuvant and inoculated into the rabbits for a third immunization. For the infestation challenge, 50 adult ticks were introduced to each rabbit for engorgement after third immunization [21]. Rabbits were bled from the marginal ear vein before immunization to obtain negative serum and were bled one week after the last immunization to obtain positive serum against rHLMLP or negative serum from the controls. The antisera from the rabbits were stored at −20 °C for later use. To evaluate the effects of rHLMLP-induced anti-tick immunity, several parameters were recorded, including the duration of feeding, engorgement weight, and mortality, and the results were analyzed by Student’s t-test.

## 3. Results

### 3.1. Nucleotide Sequence and Bioinformatics Analysis

A 938 bp cDNA sequence from HLMLP was obtained by 5′ RACE that included a complete open reading frame (ORF) and 5′ and 3′ untranslated regions. The 321 bp ORF was predicted to encode 106 amino acids with a predicted molecular mass of 11.3 kDa (Figure 1).

Subsequently, the MLP homologues from the ticks *H. qinghaiensis* (HqMLP), *Hy. anatolicum* (HaMLP), *Hy. Rufipes* (HrMLP), *B. microplus* (BmMLP), and *De. silvarum* (DsMLP) were successfully amplified using universal primers and sequenced. The sizes, predicted molecular weights, and pI values of these homologues are shown in Table 1. Analyses of the amino acid sequences of the six tick cysteine- and glycine-rich proteins revealed a highly conserved LIM domain followed by conserved glycine-rich regions (Figure 1). The LIM domain fit the pattern (CX2CX16–23HX2CX2CX2-CX16–21CX2(C/H/D)), where X denotes any amino acid, and the glycine-rich region was located immediately after the LIM domain (Figure 2). The sequence similarities between different ticks exceeded 95%, and the similarity between HLMLP and human CRP3 was 54.3%. Additionally, we constructed a 3D model structure of HLMLP based on the crystal structure of chicken CRP1 (1b8t) and calculated the electrostatic surface potential of the LIM domains in HLMLP using Swiss PDB Viewer (Figure 3). Figure 4 shows the surface maps of the LIM domains in HLMLP.

### 3.2. Phylogenetic Analysis

An MLP sequence alignment generated using BLASTX revealed high similarity with MLP from *Ixodes scapularis*. The deduced amino acid sequence of HLMLP shared 54%–99% amino acid identity with CRP3/MLP from various organisms (Figure 5). A phylogenetic tree of the CRPs was generated with the following sequences as inputs: *I. scapularis* MLP (XP002434065.1), *Amblyomma variegatum* MLP (DAA34746.1), *Caenorhabditis elegans* MLP1 (NP498300.2), *C. elegans* MLP2 (T25858), DROME MLP84B (Q24400.1), *Caligus clemensi* Mlp84B (ACO15499.1), *Drosophila silvestris* MLP84B (ACR08642.1), *Dr. melanogaster* MLP60A (ACL83203.1), *Mytilus edulis* MLP (ABB73031.1), *Lycosa singoriensis* MLP (ABX75381.1), *Gallus gallus* CRP1 (NP990579.1), *Xenopus laevis* CRP1 (NP001087798.1), *Homo sapiens* CRP1 (BAD92993.1), *Danio rerio* CRP1 (NP991130.1), *Rattus norvegicus* CRP1 (AAH62407.1), *Homo sapiens* CRP2 (NP001312.1), *Rattus norvegicus* CRP2 (Q62908.3), *Xenopus laevis* CRP2 (NP001087669.1), *Danio rerio* CRP2 (NP957191.1), *Gallus gallus* CRP2 (CAA59025.1), *Homo sapiens* CRP3 (NP003467.1), *Xenopus laevis* CRP3 (NP001079213.1), *Danio rerio* CRP3 (NP001006026.1), *Gallus gallus* CRP3 (NP001186415.1), *Rattus norvegicus* CRP3 (CAA57065.1), and human LIM-ALP (Q96JY6.1). Human LIM-ALP was used as an outlier group in the alignment of the CRP amino acid sequences. The dendrogram shows the relationships among the proteins based on the similarity of their amino acid sequences, and MLP family members from ticks formed an independent branch with other lower organisms. All vertebrate CRPs were grouped into another distinct branch in which CRP1, CRP2, and CRP3 clearly separated from each other.

### 3.3. Expression Analysis of HLMLP in Tissues and Different Developmental Stages

Total RNA was extracted from different tissues and developmental stages of *H. longicornis* and assayed by RT-PCR. The results showed that HLMLP was expressed throughout tick development (Figure 6A) and was generally upregulated in the four developmental stages. The HLMLP mRNA expression levels in the salivary glands, midgut, ovaries, and integument removed from female ticks that had fed for 4 days are shown in Figure 6B. HLMLP was more highly expressed in the integument (21.2-fold) than in the salivary glands. The relative expression of HLMLP in the midgut was 1.63-fold higher than that observed in the salivary glands. In addition, HLMLP expression in the ovaries was 1.08-fold higher than that observed in the salivary glands.

Overall, the levels of HLMLP expression in the integument were significantly higher than those observed in the midgut, salivary glands, and ovaries, and the distribution of MLP gene expression in *H. longicornis* was tissue specific.

### 3.4. HLMLP Expression in Eggs in Response to Stress

To determine whether HLMLP expression in eggs varied under temperature-induced stress, one portion of the tick eggs was maintained at 4 °C for 48 h, while the other portion was heated (25 °C for 2 h). RT-PCR analysis results showed that HLMLP expression levels were significantly upregulated by the heat treatment. When comparing the relative HLMLP expression levels at different temperatures, the results showed that HLMLP was significantly higher in eggs maintained at 4 °C than in the 25 °C treatment group (Figure 7).

### 3.5. Immunogenicity Analysis of Purified rHLMLP

The expression of rMLP was assessed by SDS-PAGE. The expected molecular weight of rMLP, including with the GST tag, was 38 kDa in a 12% polyacrylamide gel. Western blot analysis revealed strong reactivity of rabbit anti-*H. longicornis* adult serum with the rMLP-GST proteins from different ticks, while rabbit negative serum did not react with the fusion proteins. Thus, the rMLPs of different ticks showed strong cross-reactivity with rabbit anti-*H. longicornis* adult serum (Figure 8).

### 3.6. Protective Effects of rHLMLP Vaccination

The effects of rHLMLP on the physiological behavior parameters of tick feeding are shown in Table 2. No apparent differences were observed in the duration of feeding and mortality, whereas a significant difference (0.01 < *p* < 0.05) in the engorged weight was observed between the vaccinated and control groups (232 ± 12 vs. 291 ± 23 mg, respectively).

## 4. Discussion

MLPs are a group of proteins showing evolutionary conservation at the amino acid sequence level and have the primary functions of mediating protein–protein interactions and participating in the proliferation and differentiation of muscle cells [5]. MLP is specifically expressed in striated muscle and exhibits dual subcellular localization in the nucleus and cytoplasm [7]. Given the important regulatory and structural functions of MLP in myogenesis and its synergistic expression changes during myofiber-type transformation [22,23], in this study, we cloned and characterized the *H. longicornis* MLP gene as a potential candidate antigen, representing the first report on the MLP gene in ticks.

In this study, the amino acid sequences of MLPs from different tick species were analyzed. The sequences of the glycine-rich regions displayed considerable conservation in the LIM domain in ticks (Figure 2), suggesting the conservation of functional properties. This phenomenon is similar to that of the orthologous genes in humans and mice. In addition to a high degree of sequence conservation and spacing of zinc-binding residues characteristic of the LIM consensus motif, many of the non-metal coordinating residues involved in hydrogen bonding that are necessary for the establishment of the hydrophobic protein core were conserved. These residues are postulated to promote the proper overall tertiary folding of LIM domains [24]. As shown in Figure 2 and Figure 3, LIM domains were also conserved in terms of residue surface exposure and chemical surface characteristics. These hydrophobic residues are conserved or substituted by chemically similar residues in all CRP family members [4,25]. In addition, the potential nuclear targeting signal was retained, with only one conservative phenylalanine-to-tyrosine substitution observed when compared with the *Drosophila* MLP sequence (Figure 2). These results suggest that HLMLP may have a dual subcellular localization in the nucleus and the cytoplasm [7].

In the predicted phylogenetic tree (Figure 5), the MLPs of ticks clustered onto the same branch as those of other lower organisms, termed the MLP-like subset. All vertebrate CRPs diverged into three subtypes, CRP1, CRP2, and CRP3. The evolutionary distance between tick MLP and CRP3 was shorter than that between MLP and the other two subtypes. Taken together, these results suggest that the CRP protein family evolutionarily diverged from the common ancestor gene by at least two gene duplication events [26]. Some invertebrates, such as *Drosophila* and *C. elegans*, contain two members of the CRP family. However, whether there is only one MLP isoform in ticks needs to be further confirmed.

CRPs exhibit tissue-specific distribution and temporally regulated expression during embryogenesis. For example, in developing chicks, CRP1 is most prominent in tissues rich in smooth muscle, and its expression dramatically increases during smooth muscle maturation [27]. In partially fed adult ticks, HLMLP expression in the integument was significantly higher than that observed in the other tested tissues (Figure 6B). This result is in accordance with the tissue specificity of MLP gene expression in *Drosophila*, in which gene expression is spatially restricted to somatic muscles [28]. HLMLP mRNA was generally upregulated in the four developmental stages (Figure 6A), which was in accordance with the positive regulatory role of HLMLP in muscle development [7]. MLP is a novel positive regulator of myogenesis, and the expression of MLP and that of its *Drosophila* homologue, DMLP1, is enriched in striated muscle and coincides with myogenic differentiation. The transcription of HLMLP mRNA was significantly upregulated in eggs in response to thermal stress at 4 °C for 48 h. Therefore, similar to DMLP1, HLMLP may promote myogenic differentiation, with its regulation depending on the temperature of the surrounding environment. In addition, HLMLP transcription was significantly induced by blood feeding, an observation consistent with the role of MLP as a stress sensor in both cardiac and skeletal muscles [29].

The immunoreactivity of HLMLP was observed by immunoblot analysis based on the immune reaction between antisera against unfed adults and rMLP. Furthermore, strong cross-reactions between rMLPs from different species of ticks were observed (Figure 7). Thus, because MLP is highly conserved, it may be developed into a universal immunogen.

The results of the rHLMLP vaccination experiment in a rabbit model demonstrated no significant effect of vaccination on the physiological behavior of ticks, including the duration of feeding, oviposition, and egg weight per tick. One possible explanation for this observation is that MLP, as a constitutive protein, is not a suitable vaccine candidate antigen, since antisera against rHLMLP may not reach the target site or HLMLP in cells. Therefore, the expression of HLMLP in vivo was not neutralized by antiserum against rHLMLP. Another possible explanation is that HLMLP plays both structural and functional roles in the muscle of ticks but may not be essential for their survival and blood feeding.

## 5. Conclusions

This study provides novel information on the MLP gene of *H. longicornis*. The presented data provide bases for further investigation of the function of this gene and could potentially lead to a better understanding of the mechanism of myofiber determination and transformation. In addition, the results showed that the MLP is not ideal as a candidate antigen for a tick vaccine, as there was no significant effect on the development of ticks. Furthermore, these results suggest that MLP is not the best choice for screening for a tick protective antigen.

## Figures and Tables

**Figure 1 pathogens-09-00284-f001:**
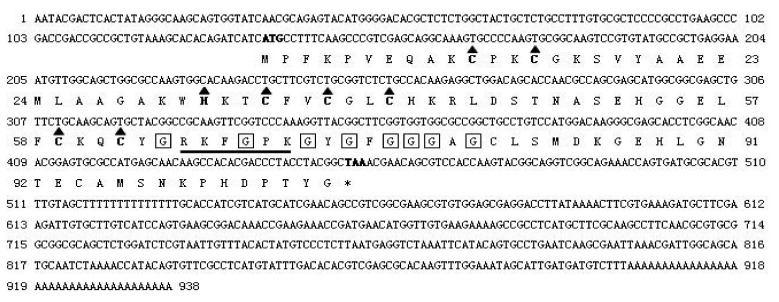
Nucleotide and deduced amino acid sequences of MLP from the tick *Haemaphysalis longicornis*. The conserved cysteine and histidine residues that define the LIM consensus sequence are marked by a triangle. Glycine residues that contribute to the glycine-rich region immediately after the LIM domain are shown in boxes. A putative nuclear targeting signal that partially overlaps the glycine-rich region is underlined.

**Figure 2 pathogens-09-00284-f002:**
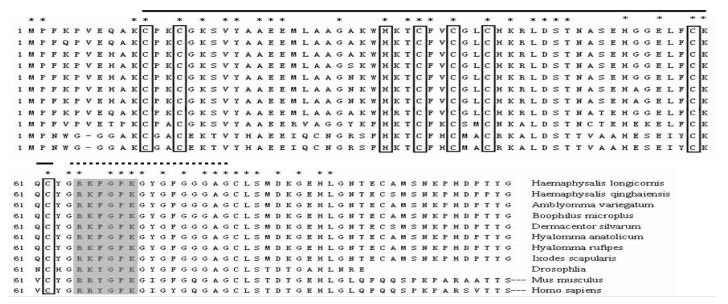
Comparison of the amino acid sequence of *Haemaphysalis longicornis* MLP (HLMLP) with the sequences of representative MLP orthologues from other ticks (*Haemaphysalis qinghaiensis* (HqMLP), *Hyalomma anatolicum* (HaMLP), *Hyalomma rufipes* (HrMLP), *Boophilus microplus* (BmMLP), *Dermacentor silvarum* (DsMLP), *Ixodes scapularis* (XP_002434065.1), and *Amblyomma variegatum* (DAA34746.1)) and those from *Drosophila melanogaster* (NP_788435.1), *Homo sapiens* (NP_003467.1), and *Mus musculus* (NP_038836.1). The conserved cysteine and histidine residues (white boxes), glycine-rich region (dotted line), LIM domain (solid line), and putative nuclear targeting signal (light grey boxes) are indicated.

**Figure 3 pathogens-09-00284-f003:**
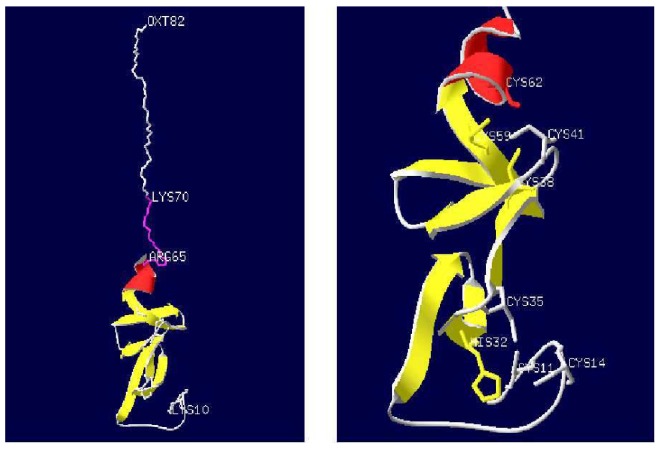
The 3D-modeled structure of MLP from the tick *Haemaphysalis longicornis*. (Left: overall structure of MLP protein; Right: the partial amino acid detailed structure folding of MLP) This model was constructed based on the crystal structure of chicken CRP2 (1b8t) using the SWISS-MODEL server. The α-helix is indicated in red, while the β-strand is indicated in yellow. The putative nuclear targeting signal (RKFGPK) is indicated in pink, and the numbers indicate the amino acid sites.

**Figure 4 pathogens-09-00284-f004:**
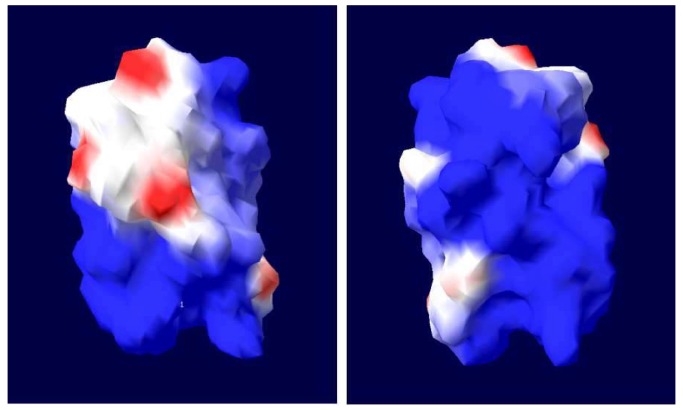
Predicted surface structures of the LIM domains in MLP (Left: Front views of LIM; Right: back views of LIM). Front and back views of the surface structures of the LIM domains are shown, revealing the differential distributions of hydrophobic residues and of the electrostatic surface potential within the domains. In the surface structures, hydrophobic, positively charged, and negatively charged residues are shown in white, blue, and red, respectively. The electrostatic surface potential was calculated using the program Swiss PDB Viewer.

**Figure 5 pathogens-09-00284-f005:**
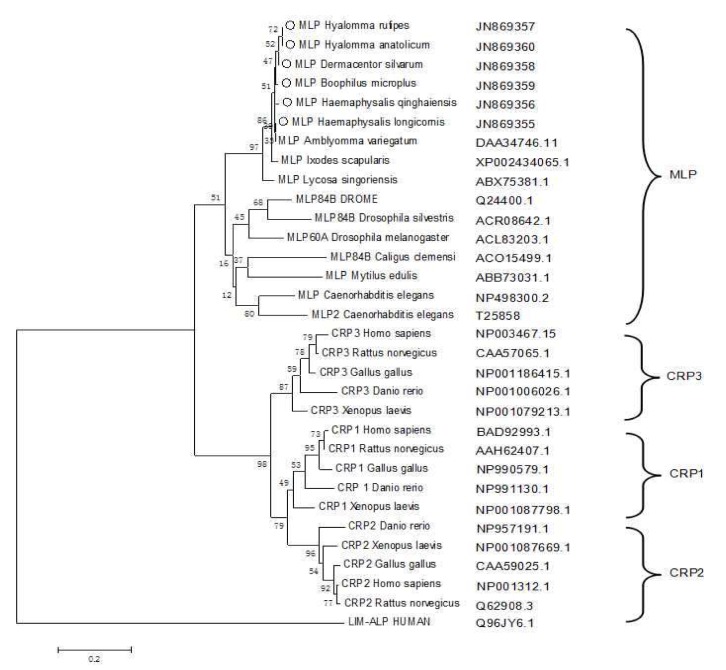
Neighbor-joining tree showing the phylogenetic relationships of the cysteine-rich protein (CRP) protein family based on protein sequences. The numbers represent the percentage of 1000 replicates (bootstrap support) for which the same branching patterns were obtained.

**Figure 6 pathogens-09-00284-f006:**
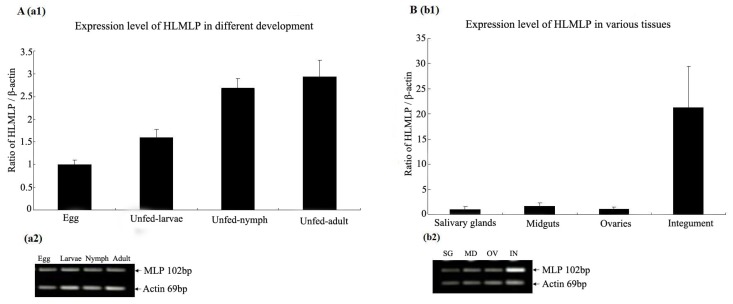
Analysis of the relative expression of MLP in the tick *Haemaphysalis longicornis* by RT-PCR. The level of HLMLP mRNA was normalized to that of tick actin mRNA. Each bar represents the average ± standard deviation, and each analysis was performed at least in triplicate. RT-PCR products were resolved in 1.0% agarose gels. (**A**)/(**a1**) The relative expression levels of HLMLP during the four developmental stages of *H. longicornis* ticks (eggs, unfed larvae, unfed nymphs, and unfed adults). (**A**)/(**a2**) The corresponding PCR products were analyzed in agarose gels. (**B**)/(**b1**) The expression levels of HLMLP mRNA in different tick tissues (ovaries, salivary glands, midgut, and integument). (**B**)/(**b2**) The corresponding PCR products of various tissues.

**Figure 7 pathogens-09-00284-f007:**
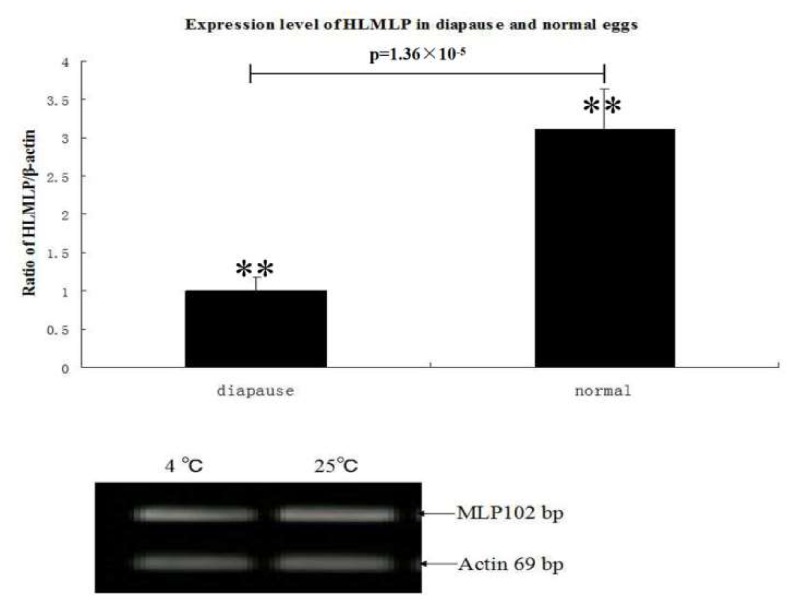
Comparative quantitative real-time PCR analysis of relative expression level of HLMLP at different temperatures, with values expressed as the means ±SD (n = 3). * The significance of the differences as determined by ANOVA analysis followed by the Student’s t-test are shown above each bar (*p* < 0.05).

**Figure 8 pathogens-09-00284-f008:**
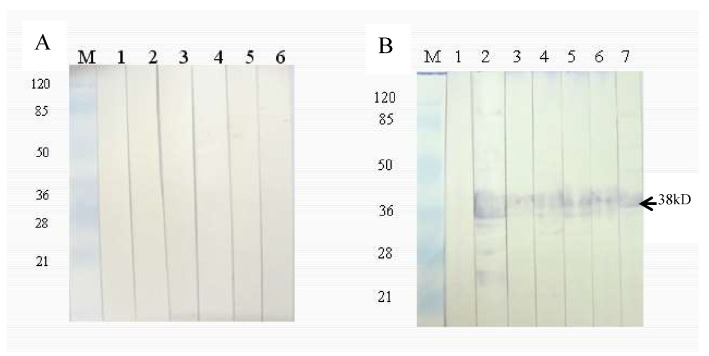
Western blot analysis of rHLMLP. (**A**) Rabbit negative serum was used as the primary antibody. Lane 1, HLMLP; lane 2, HqMLP; lane 3, BmMLP; lane 4, HrMLP; lane 5, DsMLP; lane 6, HaMLP. (**B**) Rabbit sera against *Haemaphysalis longicornis* adults were used as the primary antibody. Lane 1, GST tag protein; Lane 2, HLMLP; lane 3, HqMLP; lane 4, BmMLP; lane 5, HrMLP; lane 6, DsMLP; lane 7, HaMLP. Note: Lanes 2~7 are recombinant proteins from different ticks that showed strong cross-reactivity with rabbit anti-*H. longicornis* adult serum.

**Table 1 pathogens-09-00284-t001:** Novel muscle LIM protein (MLP) homologue genes identified in this study.

Gene	Tick Species	Accession No.	No. of Amino Acids	Molecular Weight	pI
HLMLP	*Haemaphysalis longicornis*	JN869355	106	11.3	8.58
HqMLP	*Haemaphysalis qinghaiensis*	JN869356	106	11.4	8.37
HaMLP	*Hyalomma anatolicum*	JN869360	106	11.4	8.58
HrMLP	*Hyalomma rufipes*	JN869357	106	11.4	8.58
BmMLP	*Boophilus microplus*	JN869359	106	11.4	8.58
DsMLP	*Dermacentor silvarum*	JN869358	106	11.4	8.58

Note: pI indicates isoelectric point.

**Table 2 pathogens-09-00284-t002:** The effect of vaccination with rHLMLP on tick feeding.

Parameter	Immunization Treatment	
	rHLMLP	PBS
Duration of feeding (days)	7–9	7–9
Engorged weight, mean (mg) ^a^	232 ± 12 *	291 ± 23
Mortality (%) ^b^	8 ± 2.3	10 ± 1.8
Average egg weight (mg)	47.35 ± 4.5	48.51 ± 3.8
Oviposition rate (%)	85.34	79.68

* Significance (0.01 < *p* < 0.05) was calculated using Student’s t-test. ^a^ Dead ticks were excluded; values are expressed as the mean ± standard deviation. ^b^ The mortality rate was calculated as the number of deaths during and after the feeding period and represents the total number of ticks that died on rabbits without ovipositing.

## Data Availability

The data supporting the conclusions in this study are included in the article. The MLP sequences of ticks have been deposited in the GenBank database under accession numbers JN869355 (*Haemaphysalis longicornis*), JN869356 (*Haemaphysalis qinghaiensis*), JN869357 (*Hyalomma rufipes*), JN869358 (*Dermacentor silvarum*), JN869359 (*Boophilus microplus*), and JN869360 (*Hyalomma anatolicum*).

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
