# Peer review of "Characterization of an MLP Homologue from Haemaphysalis longicornis (Acari: Ixodidae) Ticks"

_pathogens, 2020, doi:10.3390/pathogens9040284_

Round 1
Reviewer 1 Report
Muscle LIM proteins (MLP) are part of the cysteine-rich protein superfamily. In this manuscript Luo and co-workers clone a full-length cDNA from H.longicornis as well as from five other species of ticks. They then analyze expression levels in tissues, at different developmental stages and under stress. The protein was then expressed in E.coli as a GST fusion and potential protective effects of the antibody examined.
Overall the manuscript would benefit from more consistent use of language as it currently has grammatical issues or poor flow in places, in particular the introduction and results sections, while the discussion is generally very well written.
The most significant issue with the current manuscript is the strong divergence between the discussion and what is shown in the results. The discussion includes a number of points that do not appear in the results. These include:
Lines 309-310 – the amino acid sequences of the MLP protein from different tick species is not shown (a figure of an alignment would fit around lines 190-191) and hence there is no evidence shown of “the glycine-rich regions displayed considerable conservation in the LIM domain in ticks”
Lines 313-315 – nothing related to this is shown in the results
Lines 316-317 – conservation is not shown in Figure 2 or figure 3, but conservation would make much more sense than the current versions of these figures where my initial comments were “what do these figures show or tell the reader that is of any use?”
Lines 319-320 – conservation between ticks of the potential targeting signal is not shown nor is the F to Y substitution.
Lines 343-344 – HLMP transcription induction by blood feeding was not shown.
Line 351 – The effect of the vaccination on the physiological behavior was not shown.
If the manuscript had contained the data for all of these issues in the results section it would be much stronger than it currently is.
In addition:
a) The figure quality was poor in the printed version, especially for figures 1, 4 and 6.
b) The section on phylogenetic analysis has inconsistency in naming and largely repeats the figure legend.
c) No significance tests appear to have been done on any data and it is unclear if 4 significant figures for the numbers in the abstract really accurately reflects the data shown in table 2 (even if the engorged weights and mortality are significantly different). Note figure 2 legend has a “*” to show significance, but it does not appear in the table.
d) No data is shown connected with section 3.4 and the text for this section appears contradictory in places.
e) It is unclear if the antibody generated recognizes the MLP or the GST fusion which severely impacts on the interpretation of the results. Figure 6 goes some way to address this, but it is currently a poor quality composite and the background on lane 1 appears lighter than in other lanes. It is essential that this be run as a single WB
Once all changes are made, the authors should examine their conclusion to see if it still holds e.g. “biochemical and structural basis”
Minor details are missing from the methods section, including the degree of similarity (line 107), whether the GST was N- or C-terminally fused, what volume of media was used, the rpm and OD of induction, the time after the feeding period that the mortality rate was calculated over and whether it was the same for all samples.
Author Response
Response to Reviewer 1 Comments
Point 1.Overall the manuscript would benefit from more consistent use of language as it currently has grammatical issues or poor flow in places, in particular the introduction and results sections, while the discussion is generally very well written.
Response 1: Thanks for this suggestion. The language and grammatical have been revised for the manuscript.
The most significant issue with the current manuscript is the strong divergence between the discussion and what is shown in the results. The discussion includes a number of points that do not appear in the results. These include:
Point 2. Lines 309-310 – the amino acid sequences of the MLP protein from different tick species is not shown (a figure of an alignment would fit around lines 190-191) and hence there is no evidence shown of “the glycine-rich regions displayed considerable conservation in the LIM domain in ticks”
Response 2: Thanks for this suggestion. The alignment was added to the section as figure 2.
Point 3. Lines 313-315 – nothing related to this is shown in the results
Response 3: Thanks for this suggestion. The “The sequences of the glycine-rich regions displayed considerable conservation in the LIM domain in ticks” has been explain in figure 2 and the description of sequence features is also added to this section.
Point 4. Lines 316-317 – conservation is not shown in Figure 2 or figure 3, but conservation would make much more sense than the current versions of these figures where my initial comments were “what do these figures show or tell the reader that is of any use?”
Response 4: Thanks for this suggestion. The conservative feature of protein function cannot be expressed most directly in the results of homology modeling. However, the characteristics of protein conservation can be obtained through amino acid sequence. And this has already been added through figure 2.
Point 5. Lines 319-320 – conservation between ticks of the potential targeting signal is not shown nor is the F to Y substitution.
Response 5: Thanks for this suggestion. The section has been revised in figure 2, and relevant instructions are also explained and marked in the manuscript.
Point 6. Lines 343-344 – HLMP transcription induction by blood feeding was not shown.
Response 6: Thanks for this suggestion. This is a key indicator of MLP protein function for the HLMP transcription induction by blood feeding , therefore, this part of the content has been described in table2.
Point 7. Line 351 – The effect of the vaccination on the physiological behavior was not shown.
Response 7: Thanks for this suggestion. In the experiment, through the expression of the target protein, vaccines are prepared to immunize animals. The vaccines can cause a series of physiological behavior of ticks, and the data have been partially showed in table 2.
If the manuscript had contained the data for all of these issues in the results section it would be much stronger than it currently is.
In addition:
Point 8. a)The figure quality was poor in the printed version, especially for figures 1, 4 and 6.
Response 8: Thanks for this suggestion. All the figures 1, 4 and 6 will be submitted by original figures.
Point 9. b)The section on phylogenetic analysis has inconsistency in naming and largely repeats the figure legend.
Response 9: Thanks for this suggestion. The section has been revised for the figure legend and the repeats were deleted.
Point 10. c)No significance tests appear to have been done on any data and it is unclear if 4 significant figures for the numbers in the abstract really accurately reflects the data shown in table 2 (even if the engorged weights and mortality are significantly different). Note figure 2 legend has a “*” to show significance, but it does not appear in the table.
Response 10: I'm very sorry, this is a mistake for “*” missing. The above contents will be explained at "2.10. Immunization and challenge infestation", result and discussion section.
Point 11. d)No data is shown connected with section 3.4 and the text for this section appears contradictory in places.
Response 11: Thanks for this suggestion. The section has been revised for the section 3.4
Point 12. e)It is unclear if the antibody generated recognizes the MLP or the GST fusion which severely impacts on the interpretation of the results. Figure 6 goes some way to address this, but it is currently a poor quality composite and the background on lane 1 appears lighter than in other lanes. It is essential that this be run as a single WB
Response 12: Thanks for this suggestion. The figure 6 has been revised to figure 8. In the new figure these negative control have been added. And I am so sorry for the background on lane 1 appears. That is a reason, in the course of the reaction, pollution may have caused the appearance of line 1 background. But I don't think the disease affects the response of the target band. Please understand the inconvenience caused to editors or readers. The work in this area will be strengthened in the later period.
Point 13. f) Once all changes are made, the authors should examine their conclusion to see if it still holds e.g. “biochemical and structural basis”
Response 13: Thanks for this suggestion. The conclusion has been examined and revised.
Point 14. g) Minor details are missing from the methods section, including the degree of similarity (line 107), whether the GST was N- or C-terminally fused, what volume of media was used, the rpm and OD of induction, the time after the feeding period that the mortality rate was calculated over and whether it was the same for all samples.
Response 14: Thanks for this suggestion. The minor details have been revised.
Reviewer 2 Report
Review of the manuscript entitled: Characterization of an MLP homologue from 2 Haemaphysalis longicornis (Acari: Ixodidae) ticks
The manuscript is written correctly and address the molecular characterisation of MLP from Haemaphysalis longicornis. The topic is interesting for peers in the Biochemistry and physiology of invertebrate in general.
The article was previously published at the Chinese Veterinary Science / Zhongguo Shouyi Kexue 2012 Vol.42 No.8 pp.776-783 ref.20.
Comments:
In the introduction the authors should mention previous Haemaphysalis longicornis publication were the MLP genes were mention. The paper title is: De Novo RNA-seq and Functional Annotation of Haemaphysalis longicornis. Dong Ling Niu, Ya E Zhao,Ya Nan Yang, Rui Yang, Xiao Juan Gong & Li Hu. Acta Parasitologica volume 64, pages807–820(2019)
In the Western blot was used a GST tag protein as negative control. The authors should describe further molecular weight of this protein and were it should appear in the western blot. The authors should have a positive, negative control, and the amount of protein per row
Comment: H. longicornis is a three host parasite why the authors didn’t test the effect on the different stages of the H. longicornis life cycle. What adult stage was used for the rabbit infestation?
Rabbits of the first group were immunized with HLMLP, and those of the second group were immunized with PBS diluent as controls.
Question: Why don’t use as a control a group immunised with a protein not related with tick.
Vaccination of rabbits with the recombinant HLMLP conferred partial protective immunity against ticks, resulting in 20.00% mortality and a 17.44% reduction in the engorgement weight of adult ticks.
Comment: The mortality is less than 25% of the total population then the data has low significative impact as vaccine candidate for the control of tick population as well as 17.44% of reduction in the engorgement, it is really poor result.
Therefore the conclusion that this protein can be used as a future multi-antigenic vaccine candidate for the development of anti-tick vaccines is not correct.
Conclusion of the paper need to be re written because are not related with the results
Author Response
Response to Reviewer 2 Comments
Point 1. In the introduction the authors should mention previous Haemaphysalis longicornis publication were the MLP genes were mention. The paper title is: De Novo RNA-seq and Functional Annotation of Haemaphysalis longicornis. Dong Ling Niu, Ya E Zhao,Ya Nan Yang, Rui Yang, Xiao Juan Gong & Li Hu. Acta Parasitologica volume 64, pages807–820(2019)
Response 1: Thanks for this suggestion. The reference has been added as 11.
Point 2. In the Western blot was used a GST tag protein as negative control. The authors should describe further molecular weight of this protein and were it should appear in the western blot. The authors should have a positive, negative control, and the amount of protein per row
Response 2: Thanks for this suggestion. The Western blot has been revised to figure 8. In the new figure these negative control have been added.
Point 3. Comment: H. longicornis is a three host parasite why the authors didn’t test the effect on the different stages of the H. longicornis life cycle. What adult stage was used for the rabbit infestation?
Response 3: Thanks for this suggestion. In the immune challenge, There are many ticks will be used. But the the larvae and nymph ticks are small for the body, which make the statistics difficult and will cause inaccurate data, so the adult ticks were used to count various physiological indexes. Although the results are aimed at adult ticks, they are also sufficient to affect the development of the whole tick, including larvae and nymph.
Point 4. Rabbits of the first group were immunized with HLMLP, and those of the second group were immunized with PBS diluent as controls.
Question: Why don’t use as a control a group immunised with a protein not related with tick.
Response 4: Thanks for this suggestion. Here the zxskor gene from plant as a control gene was amplifified by PCR. But this gene failed when expressed in E. coli. Therefore, this section is not described here.
Point 5. Vaccination of rabbits with the recombinant HLMLP conferred partial protective immunity against ticks, resulting in 20.00% mortality and a 17.44% reduction in the engorgement weight of adult ticks.
Comment: The mortality is less than 25% of the total population then the data has low significative impact as vaccine candidate for the control of tick population as well as 17.44% of reduction in the engorgement, it is really poor result.
Response 5: The properties of a protein determines its function. This result of manuscript indicates that MLP protein is not ideal as a candidate antigen for ticks vaccine. There was no significant effect on the development of ticks. It is also suggested that the protein is not the best choice for screening tick protective antigen for other workers.
Point 6. Therefore the conclusion that this protein can be used as a future multi-antigenic vaccine candidate for the development of anti-tick vaccines is not correct.
Conclusion of the paper need to be re written because are not related with the results
Response 6: Thanks for this suggestion. The conclusion has been revised to this study provides new information on the MLP gene in H. longicornis. The data presented here provide bases for further investigation of the function of this gene and could potentially lead to a better understanding of the mechanism of myofibre determination and transformation. And the MLP protein is not ideal as a candidate antigen for ticks vaccine. There was no significant effect on the development of ticks. It is also suggested that the protein is not the best choice for screening tick protective antigen.
Round 2
Reviewer 1 Report
This manuscript is a resubmission.
While a number of changes have been made, most notably the addition of an alignment as the new Figure 2 and a new Figure 7, a large number of issues remain unaddressed.
The language is still mixed, with the changes introduced lowering the level of the language (e.g. languages issues with new text on lines 20, 24, 29, 73 ,117, 118, 119, 123, 124, 127, 165, 170, 172, 176 and 290). The amended text also lists “table 2” instead of “figure 2” on lines 316 and 326 and fails to change the figure numbers on lines 321, 340, 342 and 353. The figure legend for figure 7 is also not correct, it is a repeat of the legend from figure 6. Other language issues from the original submission are generally not addressed.
However, the bigger issue is not language related, but the separation that remains between the discussion and the results. This includes:
Lines 318-320: Where are the zinc-binding residues or the residues involved in hydrogen-bonding or those necessary for the establishment of a hydrophobic core?
Lines 348-249: Where is the data that HLMLP transcription was significantly induced by blood feeding?
Line 357: Where is the data that shows no effect of vaccination on oviposition and egg weight?
There is also a contradiction between lines 356 and 298-300 with one saying there was no effect on engorged weight, while the other says there was.
There also remain issues with the figures:
The original review commented on the poor quality of figure 1, 4 and 6 (now 1, 5 and 8) The review copy of these is still poor (authors stated they will “be submitted by original figures” in their cover letter).
It is still unclear what the authors are trying to show with figures 3 and 4. The generation of a model structure where a homologous structure is available is facile. What are these figures for? In the discussion section there is a long, generally well-written, paragraph on conservation, but these figures do not support that (or anything else). For example, the authors say that they “were also conserved in terms of residue surface exposure and chemical surface characteristics”, but figures 3 and 4 show nothing about conservation. Without this (or something else), figures 3 and 4 add nothing to the paper and do not support the discussion. There are lots of readily available bioinformatics tools to do this effectively, but even a comparison between the surface potential of HLMLP and a more distant species would greatly support the conclusions drawn “conserved in terms of….chemical surface characteristics”.
Line 262 implies that the differences seen in Figure 6 are significant and the abstract puts a P-value on that. It would greatly benefit the figure (and the new figure 7) if some indication of the significance in the differences were added e.g. ** for P-vlue < 0.05, or something similar.
As stated in the original review, it is essential that Figure 8B (was figure 6) be run as a single WB, especially given the difference in background between lanes. This has not been done.
Other issues:
There is a contradiction between lines 143 and 268 on the temperature of the heat treatment
Author Response
Point 1: While a number of changes have been made, most notably the addition of an alignment as the new Figure 2 and a new Figure 7, a large number of issues remain unaddressed.
Response 1: Thank you for this suggestion. Although the original manuscript explained some of these problems, it lacked the necessary images or tables that showed the results. Therefore, these results have been added during the revision process.
Point 2:: The language is still mixed, with the changes introduced lowering the level of the language (e.g. languages issues with new text on lines 20, 24, 29, 73 ,117, 118, 119, 123, 124, 127, 165, 170, 172, 176 and 290). The amended text also lists “table 2” instead of “figure 2” on lines 316 and 326 and fails to change the figure numbers on lines 321, 340, 342 and 353. The figure legend for figure 7 is also not correct, it is a repeat of the legend from figure 6. Other language issues from the original submission are generally not addressed.
Response 2: Thank you for this suggestion. The language has been revised by American Journal Experts (www.aje.com). In addition, “Figure 2” has been replaced with “Table 2” in lines 321 and 331, and all the figure numbers have been revised. Furthermore, the issue with the legend for Figure 7 has been corrected.
Point 3:However, the bigger issue is not language related, but the separation that remains between the discussion and the results. This includes:
Point 2:Lines 318-320: Where are the zinc-binding residues or the residues involved in hydrogen-bonding or those necessary for the establishment of a hydrophobic core?
Response 3: Thank you for this suggestion. Our results were the same as those described in reference 25, 26 and 27. Therefore, we did not include the results for the zinc-binding residues, residues involved in hydrogen-bonding or residues necessary for the establishment of the hydrophobic core. However, the relevant information has been added to the appropriate sections.
Point 4:Lines 348-249: Where is the data that HLMLP transcription was significantly induced by blood feeding?
Response 4: Thank you for this suggestion. Because our results were consistent with those described in reference 31, this passage was speculation based on the results of this reference.
Point 5:Line 357: Where is the data that shows no effect of vaccination on oviposition and egg weight?
Response 5: Thank you for this suggestion. Because there was no significant change in oviposition or egg weight in the vaccinated group, these data were not statistically analysed for significance. These missing data have been added to the revised manuscript.
Point 6:There is also a contradiction between lines 356 and 298-300 with one saying there was no effect on engorged weight, while the other says there was.section
Response 6: Thank you for this suggestion. I apologize, there was no effect on engorged weight, and this passage has been revised in lines 356.
There also remain issues with the figures:
Point 7:The original review commented on the poor quality of figure 1, 4 and 6 (now 1, 5 and 8) The review copy of these is still poor (authors stated they will “be submitted by original figures” in their cover letter).
Response 7: Thank you for this suggestion. The new figures have been uploaded.
Figure 1
Figure 5
Figure 8
Point 8:It is still unclear what the authors are trying to show with figures 3 and 4. The generation of a model structure where a homologous structure is available is facile. What are these figures for? In the discussion section there is a long, generally well-written, paragraph on conservation, but these figures do not support that (or anything else). For example, the authors say that they “were also conserved in terms of residue surface exposure and chemical surface characteristics”, but figures 3 and 4 show nothing about conservation. Without this (or something else), figures 3 and 4 add nothing to the paper and do not support the discussion. There are lots of readily available bioinformatics tools to do this effectively, but even a comparison between the surface potential of HLMLP and a more distant species would greatly support the conclusions drawn “conserved in terms of….chemical surface characteristics”.
Response 8: The 3D-modelled structure can intuitively show MLP functional sites and provide a basis for understanding MLP protein functions. These figures indicated the sites of amino acids. Although the conservation of genes cannot be shown in Figures 3 or 4, this feature was already shown in Figure 2. The amino acid numbers in Figure 3 only shows some unique sites, such as the glycine-rich region, cysteine sites, etc., and Figures 3 and 4 only show the 3D-modelled structure of MLP.
Point 9:Line 262 implies that the differences seen in Figure 6 are significant and the abstract puts a P-value on that. It would greatly benefit the figure (and the new figure 7) if some indication of the significance in the differences were added e.g. ** for P-vlue < 0.05, or something similar.
Response 9: Thank you for this suggestion. The section discussing Figure 7 has been revised, as has the legend for Figure 7.
Point 10:As stated in the original review, it is essential that Figure 8B (was figure 6) be run as a single WB, especially given the difference in background between lanes. This has not been done.
Response 10: I apologize for this issue. The background in this image is a result of contamination. However, because of the limited time for revision, it was too late to redo the experiment, but we will fix this issue as soon as possible.
Other issues:
Point 11:There is a contradiction between lines 143 and 268 on the temperature of the heat treatment
Response 11: Thank you for this suggestion. This section has been revised.

Reviewer 2 Report
The second review of the manuscript entitled
Characterization of an MLP homologue from 2 Haemaphysalis longicornis (Acari: Ixodidae) ticks
The manuscript improved respect to the previous version. However, there are English grammar errors in the text that should be attended by the authors.
Author Response
Point1: The manuscript improved respect to the previous version. However, there are English grammar errors in the text that should be attended by the authors.
Response 1: Thank you for this suggestion. The language has been revised by American Journal Experts (www.aje.com).
